

# Are sawfishes still present in Mozambique? A baseline ecological study

Ruth H. Leeney

Department of Biological Sciences, Simon Fraser University, Canada
Protect Africa's Sawfishes, Walvis Bay, Namibia

## ABSTRACT

Sawfishes (Pristidae) were formerly abundant in the western Indian Ocean, but current data on sawfish presence and distribution are lacking for most of the region. This paper summarises historical records of sawfishes in Mozambican waters and presents the findings of the first assessment of the presence and status of sawfishes in Mozambique. A countrywide baseline assessment was undertaken between May and July 2014, using interviews with artisanal, semi-industrial and industrial fishers, fish traders and fisheries monitoring staff as the primary source of information on sawfish distribution, recent catches, socio-economic value and cultural importance. Additional interviews were conducted via email or telephone with individuals running sport fishing operations or who otherwise had considerable experience interacting with the fishing sectors or the marine environment in Mozambique. Where encountered, sawfish rostra were photographed and a series of measurements and associated data were collected. In total, 200 questionnaire surveys and seven interviews with recreational fishing and dive operators were conducted, and 19 rostra were documented from museum archives and private collections, belonging to two sawfish species, the Largetooth Sawfish (*Pristis pristis)* and Green Sawfish (*P. zijsron)*. The most recent captures of sawfishes were reported to have occurred in 2014. Two key sites were identified where both recent encounters were reported and numerous Largetooth Sawfish rostra were documented. Gill nets were the fishing gear most commonly attributed to sawfish catches. Sawfishes did not hold any cultural importance in Mozambique, but they have at least some socio-economic importance to artisanal fishers, primarily through the sale of their fins. The meat did not appear to be held in high regard and was usually consumed locally. Sampling and further research is now required to confirm the presence of sawfishes and to assess the primary threats to sawfishes in those areas. At one site where a number of rostra were present and where fishers stated that they still catch sawfish, gill nets are being provided to fishers as an alternative to beach seining. This may have a serious impact on the local sawfish population and more broadly for other elasmobranchs in the area. Immediate action is required to develop a landings monitoring programme in this and other key habitats, and to encourage fishers to release sawfishes alive.

Corresponding author
Ruth H. Leeney,
ruth.leeney@gmail.com

# INTRODUCTION

Despite their former circumtropical distribution and migratory nature, little is known of current sawfish (Pristidae) distribution and abundance outside of US and Australian waters. This is of considerable concern given the critical conservation status of all five species of sawfish (*Dulvy et al., 2014*; *Harrison & Dulvy, 2014*). In particular, there is a paucity of current information on the status of sawfish populations throughout Africa, despite their historical widespread range on both the west and east coasts of the continent. Limited information suggests that most populations in African waters are now severely depleted, or have become locally extinct (*Everett et al., 2015*; *Leeney, 2015*; *Leeney & Downing, 2015*; *Leeney & Poncelet, 2013*).

At least two species of sawfish, the Largetooth Sawfish (*Pristis pristis*) and the Green Sawfish (*P. zijsron*) are reported to have occurred off the east coast of Africa (*Everett et al., 2015*); both these species are now classified as Critically Endangered on the International Union for the Conservation of Nature (IUCN) Red List (*Kyne, Carlson & Smith, 2013*; *Simpfendorfer, 2013*). In the Indo-West Pacific, Largetooth Sawfish populations are thought to have been reduced by 80% or more, based on a reduction in extent of occurrence over a period of three generations (i.e., 1969 to present; *Kyne, Carlson & Smith, 2013*). The toothed rostrum and demersal habits of all sawfish species make them extremely susceptible to capture in gillnets and demersal trawl nets (*Simpfendorfer, 2013*; *Kyne, Carlson & Smith, 2013*).

In Australia, adult Largetooth Sawfish inhabit marine and estuarine environments and juveniles are found in rivers and freshwater regions of estuaries (*Peverell, 2005*; *Thorburn et al., 2007*; *Whitty et al., 2009*), whilst Green Sawfish inhabit mostly inshore areas, including estuaries and river mouths, and are strongly associated with mudflats and mangroves (*Stevens, Pillans & Salini, 2005*; *Stevens et al., 2008*; *Phillips et al., 2011*). In The Gambia, juvenile Largetooth Sawfish were caught in the Gambia River in the 1970s, suggesting the river was an important pupping and nursery ground (*Leeney & Downing, 2015*). Rivers and estuaries are clearly important habitats for both species, and mangrove systems also appear to indicate good habitat for Largetooth Sawfish (*Fernandez-Carvalho et al., 2013*). The mouths of major rivers and areas of extensive mangrove cover can thus be used as focal points for data collection in regions with little current knowledge of sawfish distribution.

Mozambique's extensive coastline and numerous river systems are host to many industrial, semi-industrial and artisanal fisheries. The shallow water shrimp fishery is one of the largest of the industrial fisheries and crustaceans are also targeted in deeper waters (*Kiszka & Van der Elst, 2015*). The prawn trawling industry contributes substantially to GDP and also provides significant amounts of foreign capital (*Fennessy et al., 2008*). The main trawling areas are the expansive (50,000 km$^2$) Sofala Bank grounds along the north of the coast (off the provinces of Sofala and Zambezia and the southern half of Nampula province), and smaller grounds off the Limpopo River and in Maputo Bay (*Fennessy & Everett, 2015*). Semi-industrial gillnet fisheries targeting sharks were established in the late 1990s, operating mainly in Maputo Bay and Inhambane Bay and targeting primarily coastal or shelf-associated species (*Sousa, Marshall & Smale, 1997*). Demersal gill net fisheries for deep water squalids still operate in Mozambican waters (*Kiszka & Van der Elst, 2015*).

Mozambique also issues foreign fishing rights for tuna and other large pelagics, but little or no information on catch is received from these foreign operators (*Van der Elst et al., 2010*). Of all the industrial and semi-industrial fisheries, the trawl fishery is thought to catch the most significant quantities of elasmobranchs, as bycatch (*Kiszka & Van der Elst, 2015*).

Artisanal fisheries are extensive along the coast and in Mozambique's numerous river systems. According to census data collected by the Institute for the Development of Small-Scale Fisheries (IDPPE), there were approximately 128,044 artisanal fishers without vessels (e.g., those engaged in fishing from the shore, diving, collecting shellfish), 157,465 working from vessels and 39,550 artisanal fishing vessels operating in 1,586 centres of fishing activity nationwide (including both freshwater and marine areas) in 2012 (*Ministério das Pescas, 2013*). This is likely an underestimate, since fishers in the north report that Tanzanian fishers also cross the border and fish in Mozambican waters, and because many fishers are based in remote areas which may not have been included in the census (RH Leeney, pers. obs., 2014–2015). Gears used in small-scale fisheries include beach seines, surface and bottom-set gillnets, longlines, hand lines and weirs (*Everett et al., 2013*). Some specialised artisanal fishers target sharks (RH Leeney, pers. obs., 2014–2015), and short-term opportunistic targeting of specific elasmobranch species may occur in certain areas (*Kiszka & Van der Elst, 2015*). According to the FAO FishStat database, Mozambique landed an average of 572.7 tons of shark product annually between 2003 and 2012, exclusively from the WIO. Although Mozambique did not report any exports of any shark products between 2003 and 2013, a number of countries reported that they had imported shark products from Mozambique in this time frame, primarily frozen shark meat (data sourced from UN Comtrade and Eurostat).

A recent conservation strategy released by the IUCN (*Harrison & Dulvy, 2014*) highlighted the urgent need for baseline data on sawfishes throughout much of their historical range. There have been no comprehensive assessments of the current status of sawfishes in east Africa and as such, their status in Mozambique, including current abundance, any contractions in range, historical declines and local threats, is completely unknown. This study provides the first historical and current account of sawfishes in Mozambique, presenting collated historical records alongside recent interview data to elucidate former key habitats for sawfishes, areas where they may still be encountered, local threats to sawfishes, and the socio-economic and cultural importance of sawfishes to fishing communities. This baseline research has highlighted areas where sawfish populations are likely to persist and where further research will be required in order to better understand the ecology of these populations and the threats they face. Recommendations are made for future research and management activities likely to be feasible and successful, given the challenges conservation projects face in this region.

## METHODS

### Study area

The Mozambican coastline extends 2,770 km along the south-eastern edge of the African continent and can be divided into three main areas: the dune coast (Delagoa Bight) in the

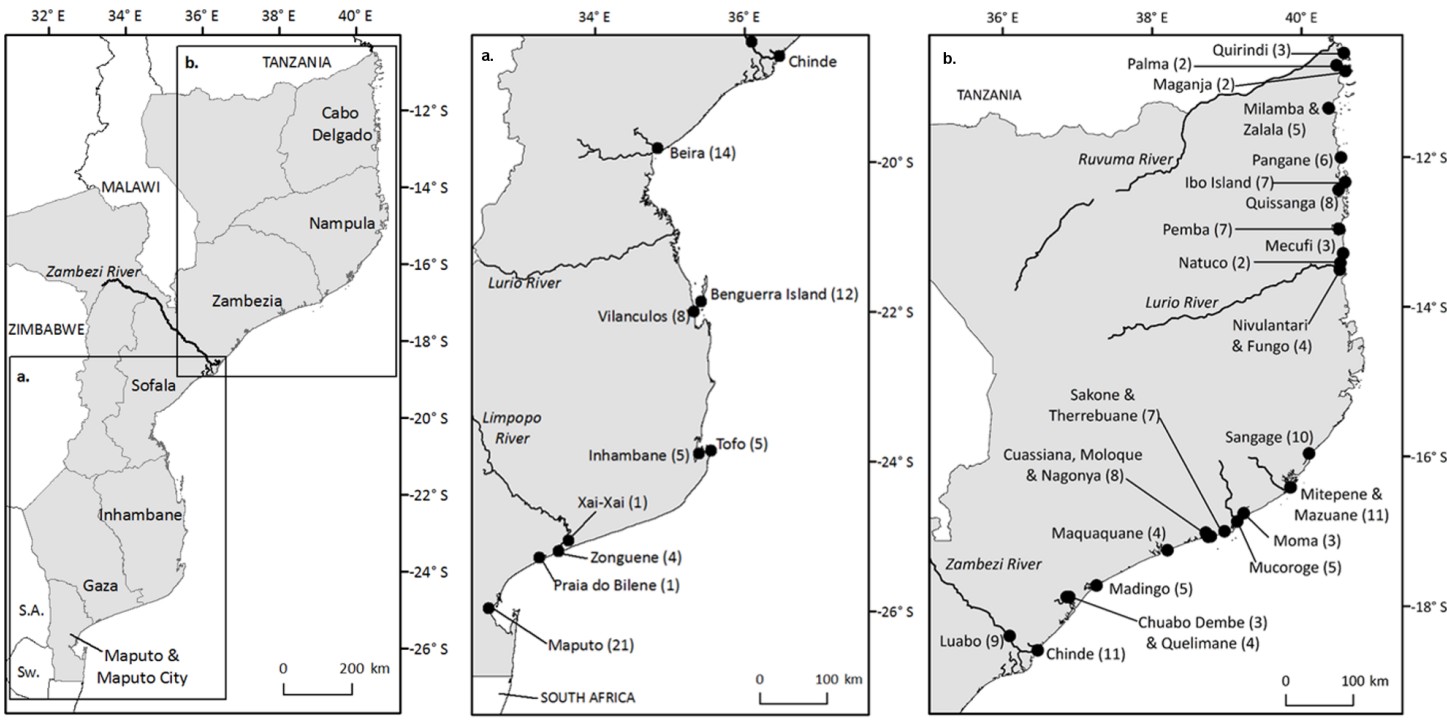

**Figure 1** **Map of interview sites.** Interview sites in (A) southern and (B) northern Mozambique, with number of interviews completed at each site, in parentheses.

south, the swamp coast (Sofala Bank) in the centre and the coral coast (São Lazaro Bank) in the north. Mozambique's Exclusive Economic Zone has an area of 562,000 km² (*Pierce et al., 2008*). Major rivers punctuating the coastline include (from north to south) the Rovuma, Lurio, Meluli, Zambezi, Save and Limpopo; the mouths of these rivers formed the basis of initial site selection for the interview surveys conducted for this study (Fig. 1). However, much of the northern portion of Mozambique's coastline (north of Vilankulos, Inhambane province) is accessible only using off-road vehicles and local fishing boats, and thus the availability of suitable vehicles and local staff determined to a large extent the sites visited.

## Historical records of sawfishes in Mozambique

There are a number of historical records of sawfishes occurring in Mozambique, many of which provide insight into interactions between local human populations and sawfishes. A literature search was carried out to collate evidence of sawfish occurrence in Mozambique as documented by writers, explorers and biologists. Whilst such reports likely do not exhaustively document all historical sawfish habitats, such information often provides the only available clues in the search for existing sawfish populations and can thus prove a useful starting point for baseline studies. Records of sawfish rostra in museum collections were also included, as were all records of sawfishes in Mozambican waters from the International Sawfish Encounter Database (ISED).

**Table 1  Primary professions of interviewees ($n = 200$).** Most artisanal fishers used more than one gear type.

| Interviewee's profession | $n$ |
|---|---|
| Fisher—industrial trawler/semi-ind[a] | 24 |
| Fisher—artisanal[b] | 160 |
| IIP monitoring staff | 11 |
| Seller/processor | 5 |

**Notes.**
[a] Includes trawlers, semi-industrial longliners, semi-industrial 'arrachte'.
[b] Includes handline, gillnet, beach seine, artisanal trawl, weir, trap, diving/spearfishing, collecting shellfish.

## Interview surveys

Research was conducted with the permission of the director of the Mozambican Institute for Fisheries Research (Instituto Nacional de Investigação Pesceira, hereafter IIP) in Maputo. Interviews were carried out at 38 sites along the Mozambican coast (Table 1 and Fig. 1). These sites included towns and villages, fish landings sites and ports. All interviews were carried out by the author, in most cases accompanied by a staff member from IIP. In Nampula province, the author was accompanied by a staff member from the NGO alliance WWF-CARE, and for sites along the Zambezi River, by a member of staff from Eduardo Mondlane University. In some areas, IIP did not have a presence but the IDPPE worked closely with communities. In these areas, we first sought permission from the local IDPPE team to work in the area and in most cases were accompanied by an IDPPE representative. Artisanal, semi-industrial and industrial fishers were interviewed, as well as several fish traders and processors who worked closely with fishermen and thus similarly had extensive knowledge of local fish catches. Several IIP staff members who worked as fisheries observers on industrial and semi-industrial fishing vessels were also interviewed.

Interviews were conducted either in Portuguese or in the local language used by each interviewee. Interviews were conducted according to the availability of each fisherman when approached by the interview team and took place throughout the day, either at landing sites and harbours, fisheries cooperative buildings or in the respondent's residence. We first introduced ourselves and explained that the research was being conducted in collaboration with IIP. We explained to each interviewee that we were collecting information about rare marine species in Mozambique, but did not specifically mention sawfishes. We assured each interviewee that the questionnaire was anonymous and likewise explained that the respondent was not obliged to answer any question s/he did not wish to. Permission to conduct the interview was requested verbally from each interviewee prior to starting the interview. If the interviewee agreed to participate in the interview, he or she was then shown a high-quality colour photograph of a sawfish, and was asked whether s/he recognised the fish and had ever seen one before. If the respondent could not identify the sawfish, the interviewer collected only basic data relating to the individual's age, job, number of years of experience in fisheries and (for fishers) the types of fishing gear used. If the respondent could identify the sawfish, the full interview was conducted (Appendix I).

An internet search was also conducted for sport fishing operators, dive operators and resorts which offered fishing or diving to guests. An email was sent to each operator to

explain the aim of the research project and to request as much information as possible on past or current observations of sawfishes, including the nature (rostrum only/full body (dead)/live specimen), date and location of each encounter. The information provided by these individuals was not collected through a formal interview structure and thus has been presented separately.

## Sawfish rostra

Enquiries were made as to the availability of sawfish rostra, both during interviews and also during visits to the Xipamanine traditional medicine market (Maputo), the National Museum of Natural History in Maputo, the Museum of Natural History on Inhaca Island and fish markets throughout the country. A series of standard measurements were taken from all rostra encountered. Each rostrum was assigned a species identification using the number of rostral teeth and the ratio of standard rostral width (SRW) to standard rostral length (SRL). Rostral tooth count and SRW:SRL ratio was calculated for each rostrum, and was compared to published data on the known ranges of these metrics for each species (*Whitty et al., 2013*).

## Analysis of interview data

Of 200 interviewees, 25% (50 individuals) either did not know what a sawfish was, or had heard of them but had never seen one. Only information from respondents who were familiar with sawfishes and had observed one at least once during their lifetimes was used to describe the trends presented in the Results section (also see Caveats section, below, regarding information from industrial and semi-industrial fishermen and fisheries observers). The dates of respondents' last observations and catches of sawfishes were binned by decade to provide insight into whether sawfish observations are still a common occurrence. To assess whether certain gear types were more frequently responsible for catching sawfishes, information was summarised on the type of gear used to catch the sawfish most recently observed by each interviewee (excluding observations from industrial and semi-industrial fishers and fisheries observers). Fishers' perceptions regarding changes in sawfish abundance and the causes for those changes were described, but a limited number of responses prevented any detailed analysis. Likewise, the stated traditional and current uses of sawfish products have been described. Information on where respondents had observed or caught sawfishes is perhaps the most important output from this research, as it can be used to target areas for future research and conservation efforts. However, both Green and Largetooth Sawfish, listed as Critically Endangered, are potentially of high economic value and are threatened by trade; the current distribution of these species in Mozambique is not well documented. IUCN recommendations[1] on reporting the distribution of Endangered and Critically Endangered species have thus been followed and the key sites identified during this study have not been named here.

## Caveats

The caveats associated with using interviews to collect data have been discussed in detail in *Leeney & Poncelet (2013)*, and may include interviewees withholding information if they fear repercussions for any interactions they report with the species of interest, or if they disagree with perceived or stated conservation goals (*Silver & Campbell, 2005*; *Le Douget,*

[1] IUCN Red List. 2012. Rules of Procedure IUCN Red List Assessment Process 2013–2016—Annex VI. http://www.iucnredlist.org/documents/Rules_of_Procedure_for_Red_List_2013-2016.pdf.

*2009*). Previous studies of this type have used local interviewers in an attempt to reduce feelings of mistrust or fear (e.g., *Leeney & Poncelet, 2013*). Such a strategy was not possible during this study, due to the limited time available in which to conduct interviews and the extent of the Mozambican coast, which required the use of multiple local collaborators and prevented the mobilisation of a few well-trained interview teams. The author was accompanied at all times by a Mozambican collaborator and where possible, by a local fisheries or NGO officer known to the interviewees, and an emphasis was placed on ensuring that interviews were conducted in an informal and cordial manner. Nonetheless, it is possible that some interviewees may have withheld information. Imperfections in memory also cause inaccuracies in the information collected (*McKelvey, Aubry & Schwartz, 2008*). Due to these factors, local ecological knowledge (LEK) is not a substitute for ecological surveys, and the resulting data should thus be analysed accordingly—as valuable but imprecise information which can provide insight into species presence or absence and trends in local abundance over time. The value of interview surveys is primarily as a cost-effective and time-effective means of collecting basic information from a large geographical area, and is especially useful when no current baseline data exist and when the species is cryptic or is believed to be rare. The initial, largely anecdotal information collected then allows researchers to pinpoint areas deserving of further research effort.

Photographs taken by an IIP observer in December 2014 (after the completion of the interview study) whilst on a vessel trawling approximately 10 km east of Inhaca Island (Maputo Bay) and sent to the author revealed that the bycatch comprised several saw sharks (Pristiophoridae) but no sawfishes. The same fisheries observer who collected the photographs had previously seen the image of a sawfish used during this study, and had mis-identified the saw sharks he photographed as sawfishes. It is thus possible that some of the reports from both fisheries observers and industrial or semi-industrial fishers pertain to saw sharks rather than sawfishes. Likewise, an IIP observer based in Maputo and regularly stationed on semi-industrial long-liners and trawlers stated that he saw small 'sawfish' of c. 0.5 m in length, about once every three fishing trips, and that they were usually thrown back. The size of these animals and the frequency with which they are caught suggest that these fisheries are regularly catching saw sharks. For this reason, the data from interviews with fishers working on semi-industrial and industrial vessels and from IIP observers ($n = 35$) have been presented separately and should be interpreted with caution.

## RESULTS

### Historical records of sawfishes in Mozambique

Historical reports of sawfishes have been compiled in Appendix II. The earliest available record of sawfishes in Mozambique was made by Livingstone during his exploration of the Zambezi River, between 1858 and 1864 (*Livingstone & Livingstone, 1866*). Many historical records came from the Zambezi (e.g., *Boulenger, 1909*; *Wallace, 1967*) but sawfishes were also reported to occur in the Save River (*Smith, 1950*; *Smith, 1952*; *Jubb, 1961*) and the Shire River (a tributary of the Zambezi; *Swann, 1910*) as well as in Delagoa Bay (now known as Maputo Bay) and in the vicinity of Maputo (*Smith, 1950*; *Jubb, 1961*). *Tanser (1975)*
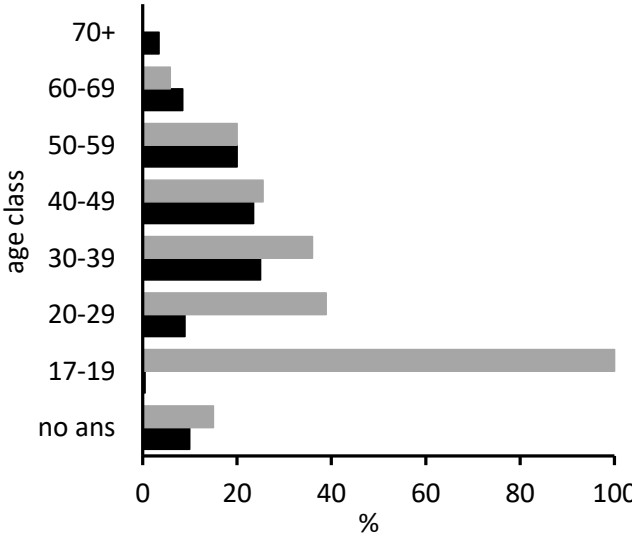

**Figure 2** Age distribution of respondents (black bars) as proportions of the total number of intervie-wees ($n = 200$) and age distribution of the subset who did not recognise or had never seen a sawfish (grey bars), as proportions of the number of individuals in the corresponding age category ($n = 50$). 'No ans' indicates respondents who did not provide their age ($n = 20$).

noted that '*sawfish are occasionally caught at the confluence of the Lundi and Sabi rivers*'. The Lundi River (now known as the Rundi River) is a tributary of the Save (Sabi) River and the two join in the south-eastern corner of Zimbabwe, just before the Save crosses the border into Mozambique. In addition to these reports, four sawfish rostra known to be of Mozambican origin are held in museum collections (British Natural History Museum; Museum für Naturkunde, Berlin and Museu da Ciência, Universidade de Coimbra); all are specimens of *P. pristis* (Appendix I). This information suggests that at least in the last century, sawfishes were relatively abundant in Mozambican waters.

## Interview surveys

Between 28 May and 04 August, 2014, 200 structured interviews were carried out at 38 sites along the Mozambican coast (Table 1, Fig. 1 and Appendix III). Of these, 199 interviews were with individuals and one was carried out with a group of fishers. All interviewees were male as no female fishers were encountered during the study. Of the 200 interviewees, 160 were artisanal fishers, using a wide variety of gears and methods including handline, bottom gill net, surface gill net, beach seine, diving/spearfishing and collecting shellfish, whilst five individuals worked directly with artisanal fishers, as vendors or processors. Interviewees also included industrial or semi-industrial fishers (24 individuals) and IIP observers or monitoring staff (11). In addition, 7 non-structured interviews were conducted by phone, email or in person with diving or sport fishing operators. The information from these non-structured interviews is presented separately.

The age distribution of all interviewees ($n = 200$) and of the sub-group of respondents who did not recognise the image of a sawfish or had never seen one (as proportions of the number of respondents in each corresponding age class; $n = 50$) is presented in Fig. 2.

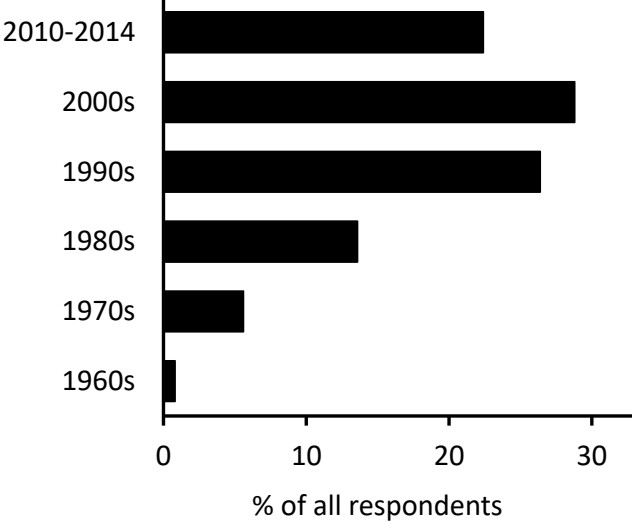

**Figure 3  Most recent sightings of sawfishes.** Most recent sightings and catches of sawfishes (binned by decade) reported by all interviewees (except IIP staff, industrial and semi-industrial fishers) who had seen a sawfish at least once. Percentages presented represent the number of responses in each decade as a proportion of all interviewees ($n = 125$); 3 interviewees did not provide a response.

The majority of respondents (69%) were between 30 and 59 years old. The sub-set of interviewees who had never seen a sawfish included individuals of almost all age categories, but made up greater proportions of younger age classes (20–39 years old).

Of all respondents ($n = 200$; including IIP observers, semi-industrial and industrial fishers), 75% ($n = 150$) stated that they recognised the image of a sawfish and had seen a sawfish at least once (but see notes in Methods regarding possible misidentification).

## Local names

The local names for sawfish varied amongst regions; the names noted in each province are listed in Appendix IV. In Zambezia province, some interviewees stated that the local name for sawfish was '*mokuru haji*', but several of these individuals later corrected themselves and clarified that this name referred to '*the same fish*' (pointing to the sawfish image), '*but without a saw*'—most likely referring to guitarfish (Rhinobatidae). Swahili is spoken in northern Mozambique and the local name recorded in Cabo Delgado province, *papa panga*, translates as 'knife shark'.

## Interview data from artisanal fishers, fish vendors and processors ($n = 165$)

### Most recent sawfish encounters

Of 165 interviewees involved in artisanal fisheries (as a fisher, processor or seller), 76% ($n = 125$) had seen a sawfish at least once during their lifetime. The dates of respondents' most recent sawfish sightings and captures ranged from 1968 to 'last week' (indicating July 2014). Respondents who said their last observation occurred between 2000 and 2009 accounted for 29% or all interviewees, whilst 22% said they last saw a sawfish between 2010 and 2014 (Fig. 3).

Of the 14 reports of sawfish captures and sightings stated to have occurred in 2013 and 2014 (up to the time of the study), 8 were reported to have occurred in Zambezia province, three in Inhambane province, two in Nampula province and one in Cabo Delgado. In a number of villages in Zambezia and Nampula provinces where individuals who reported recent catches of sawfishes were interviewed, sawfish rostra were also documented ($n = 8$), providing additional evidence to support the presence of sawfishes in these areas. One of the rostra owned by a fisherman in Zambezia province appeared to be from a relatively recent catch (the tissue at the base of the rostrum was not completely dried). In June 2016, a fisherman who had been interviewed in 2014 was contacted during a follow-up visit to a village in Zambezia province. He reported that he had caught an adult sawfish in 2014, several months after being interviewed, and he had retained the rostrum to show the author. The rostrum measured 1.09 m (SRL) and was identified as belonging to a Largetooth Sawfish. For two other recent catches, reported to have occurred in June 2014 in Zambezia province and July 2014 in Nampula province, the rostra had been retained and were provided to the interview team. Both rostra belonged to Largetooth Sawfish. For the other observations reported to have occurred in 2013/2014, the mean estimated total length of the sawfishes observed was 2.3 m (range: 1–5 m; $n = 12$). The most recent sawfish captures were stated to have occurred the week prior to interviews conducted on the 23 and 27 June 2014.

### Gear types with which sawfishes were caught

Interviewees were asked which type of fishing gear had been used to catch the last sawfish they had seen. The majority of respondents (52% of 125 interviewees) stated that various types of gill net, including deep-water gill nets, surface gill nets and 'shark nets' (a large mesh gill net), had been used (Fig. 4). The second most commonly-cited gear type was lines (which included both hand lines and longlines; 11%), followed by trawl nets (8%). Sawfishes also appear to have been caught by a diverse range of other gear types including weirs, beach seines and spears, but in far fewer instances.

### Perceived changes in sawfish abundance

Of 125 respondents who stated that they had observed sawfishes, 30% ($n = 37$) had seen a sawfish only once. Respondents who had observed more than one sawfish during their lifetime ($n = 88$) were asked whether they had noted a change in sawfish abundance over time. In general, however, the question was not well understood and only 32 individuals provided a response. The majority ($n = 27$) stated or indirectly indicated (by discussing possible causes for a decrease in sawfish abundance) that sawfish numbers locally had declined over the course of their lifetime; two individuals believed that sawfishes had always been rare and three interviewees stated that sawfishes could 'still be found'. Amongst those who stated that sawfishes had declined and who suggested one or more causes for this decline ($n = 18$), an increase in the number of fishers and the amount of fishing gear in the water was the most frequently-mentioned. Other reasons for declines in sawfish encounters included sawfishes moving offshore to avoid motor noise, fuel in the water and industrial fishing vessels; loss of mangrove cover; fishers no longer using the gill nets which most effectively catch sawfishes; younger fishermen not targeting sawfishes, and sawfishes not reproducing adequately to maintain their populations.

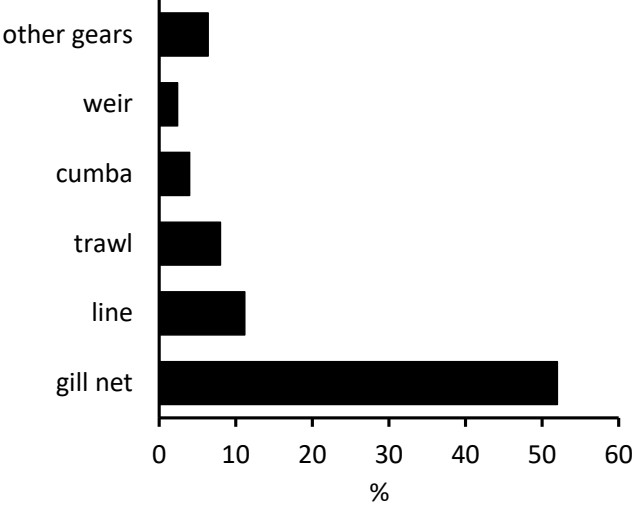

**Figure 4** **Gear types used to catch sawfishes.** The gear types with which sawfish were caught in the case of each interviewee's most recent sawfish sighting (responses from artisanal fishers, processors and vendors only). Percentages presented represent the number of responses in each response category as a proportion of all interviewees ($n = 125$); 18 respondents didn't know the gear type used and 3 said they had only seen live sawfishes. *Gill net* includes the responses 'emalhar de fundo' (deep-water gill net), 'emalhar superficie' (surface gill net), 'rede de tubarão' (shark net) and 'jarifa' (gill net). *Line* includes 'palangre' (longlines) and hand lines. *Trawl* refers to 'arrasto' (artisanal trawl net). *Weir* refers to structures locally called 'gamboa', *Other* includes marlin rod and line, spear, beach seine, 'armadilha' (trap) and 'chicocorta' (mosquito net traps).

## Distribution

At least two key regions were identified where it appears likely that sawfishes are still present, based on reports of recent catches and the presence of numerous rostra. This information has been provided to the Mozambican Institute for Fisheries Research (IIP) and is also held by the IUCN Shark Specialist Group and the International Sawfish Encounter Database.

Non-structured interviews were carried out with five sport fishers, a dive operator and an individual who conducted frequent dives around the Primeiras and Segundas Archipelago between 2010 and 2011, which provided further insight into the historical distribution of sawfishes. Sawfishes were reportedly caught in shrimp nets on the southern side of Maputo Bay, 'years back' (R Jacobs, pers. comm., 2014). Between 1990 and 1993, sawfish rostra were observed washing up on the north point beach of Bazaruto Island (L Erasmus, pers. comm., 2014). Mr. Erasmus continues to fish from the beaches of Bazaruto and catches several shark species including blacktip reef (*Carcharhinus melanopterus*), Zambezi (*C. leucas*) and hammerhead (*Sphyrna* sp.) sharks, but has never caught a sawfish. A sport fishing operator based in Vilankulos since 2000 and fishing around the Bazaruto Archipelago had never caught a sawfish, but knew of one sawfish catch in the area, landed by local fishers on Benguerra Island, around 2007 (M O'Kennedy, pers. comm., 2014). Sawfishes had been observed in Zambezia province as recently as 2011 (H Vosloo, pers. comm., 2014).
[2]Based on an exchange rate of MZN 1 = US$0.01305 for 07 Nov 2016.

### Cultural and socio-economic importance and uses of sawfishes

Interviewees who had seen a sawfish at least once were asked about the uses to which they or other fishers had put the meat, fins and rostra of the sawfish they most recently caught or observed being landed. Of 125 interviewees who had seen a sawfish at least once, 17 individuals did not know of any uses of sawfishes or did not provide a response, whilst 108 individuals provided responses regarding the uses of various parts of sawfishes. The most commonly-stated uses of captured sawfishes were direct consumption (57%) and sale of the meat (50%; Fig. 5). Forty-five percent of respondents mentioned that sawfish fins were sold, whilst 11% stated that sawfish fins had been discarded. Fins were usually sold to 'collectors' who probably act as middlemen between fishers and exporters. The sale prices quoted for sawfish fins were far greater than those for the meat, and the latter appeared to be sold only for local consumption. The sale prices quoted by interviewees who reported having caught or observed the catch of a sawfish between 2004 and 2014 varied considerably, and several interviewees noted that the value of fins was not as great as it had been some years previously. Sawfish meat was stated to fetch between 5 and 50 MZN (US$0.06–0.65[2]) per 'piece' (the standard size of a piece of meat varies amongst markets but is likely to be between 0.5 and 1 kg), and three interviewees stated that whole sawfishes had sold for 500–600 MZN (US$6.50–7.80; the sizes of the fish were not specified). The prices for fins depended on their size, and fins appeared to be sold both individually and by weight. Per-fin prices ranged from 50 to 1200 MZN (US$0.65–15.00), depending on size. By weight, prices for sawfish fins ranged from 45 to 7000 MZN/kg (up to US$91/kg), with this maximum price stated to have been paid in 2009. An interviewee who had caught a sawfish in 2014 estimated that sawfish fins were worth 400 MZN/kg, whilst guitarfish fins were worth 800 MZN/kg. Guitarfish fins were stated by another interviewee to be the most valuable, selling for 4,500 MZN/kg (US$58).

Sawfish rostra were most commonly discarded (38%), whilst 18% of respondents stated that they were 'kept', usually as a decorative item in a house, or displayed at a fishing camp. The sale of rostra was mentioned by only 6% of respondents (Fig. 5). Several interviewees stated that the meat is not highly regarded and thus there is not a substantial demand for it. Other uses of sawfish mentioned by interviewees included grinding the rostral teeth to use as a garden fertiliser (2 respondents), as a source of oil for cooking (1 respondent) or as a waterproofing agent for fishing boats (1 respondent); using the rostral teeth as needles in the construction of grass mats (1 respondent) and hanging clothes from the rostrum (1 respondent).

When asked whether they knew of any beliefs or cultural practices associated with sawfishes or any part thereof in their village, the majority of interviewees appeared not to understand the question or responded that sawfish were primarily a source of food. A fisherman in Chuabo Dembe, Zambezia province, mentioned a traditional magical practice involved grinding the rostral teeth and mixing them with plants and other ingredients to make a paste, which would be spread on a fisherman's net to ensure a good catch. Another interviewee stated that sawfishes 'saved' fishermen, presumably if their boats capsized.

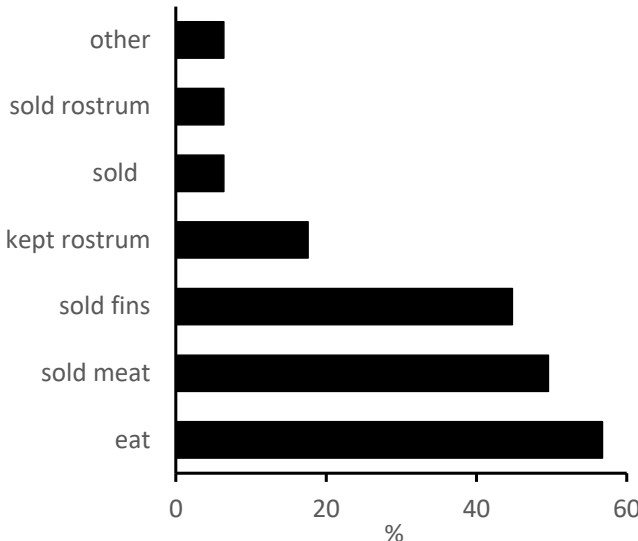

**Figure 5 The uses of sawfishes by artisanal fishers and their communities.** Responses pertained largely to the way in which the sawfish that each respondent had most recently observed had been used. Percentages presented represent the number of responses in each response category as a proportion of all interviewees ($n = 125$); 16 individuals did not provide a response but most respondents provided more than one answer. *Eat*, consumed directly by the fisher or his family/ community; *sold meat*, sawfish meat was sold; *sold fins*, sawfish fins were removed and sold separately; *kept rostrum*, interviewee stated that rostrum was either retained (no further details provided) or used for decorative purposes in either a house or at a fishing camp; *sold*, sawfish was sold whole, or the interviewee did not specify which part of the sawfish was sold; *sold rostrum*, rostrum of sawfish was sold to a collector; *other*, includes use of oil as waterproofing agent or for cooking; use of rostrum as hooks; use of rostral teeth as fertiliser and as needles in the construction of grass mats.

## Interview data from IIP observers, industrial and semi-industrial fishers ($n = 35$)

Of the 24 interviewees working on industrial or semi-industrial vessels, 18 were involved with trawl fisheries while the rest were employed in long-line fisheries. Only one of the six long-line fishers had ever encountered a sawfish. Of those involved in trawl fisheries, 15 stated that they had encountered sawfishes, and two stated that their most recent observation had been only several weeks previously (but see caveats, below). In addition, seven IIP fisheries observers, with experience of between 5 and 35 years on industrial or semi-industrial fishing vessels, stated that they had seen sawfishes during their time on those vessels. Five of the 6 observers stated that they had seen sawfishes caught by trawl nets; the other that he had observed a sawfish caught by the semi-industrial line fishery. One IIP observer stated that 'small sawfish' were often caught in the industrial offshore shrimp fishery (operating in 3–400 m depth) in southern Mozambique. However, as noted in the Methods section, it is likely that at least some of these observations pertain to saw sharks rather than sawfish, especially given the bycatch of several saw sharks by the deep water trawl fishery off Inhaca Island in 2014. Nonetheless, a fisher interviewed in Beira owned a sawfish rostrum which was taken from an animal caught during shrimp trawling activities on the Sofala Bank (the fisher himself had been a part of the crew). The rostrum, from

a Green Sawfish, was documented as part of this study (Appendix V) and provides some evidence that sawfishes have also been caught by semi-industrial and industrial vessels.

### Sawfish rostra

No sawfish rostra were found during a visit to the traditional medicine section of Xipamanine market (Maputo) in July 2014, and vendors stated that they did not usually have rostra for sale. Enquiries were made regarding sawfish rostra and more generally about shark fins, with a shark fin vendor in Quelimane, but the vendor stated that he no longer traded these products. The Natural History Museum on Inhaca Island did not contain any sawfish rostra. The collection of the National Museum of Natural History in Maputo contains one rostrum from a Largetooth Sawfish, as well as taxidermy specimens of three Largetooth Sawfish (of which measurement data were collected from only one, as the other two have been considerably altered for display) and one Green Sawfish. However, no information is held on the date or location of capture of these specimens.

Including the specimens in the National Museum of Natural History, a total of 14 sawfish rostra were recorded in Mozambique for this study: 12 in 2014 and two more on a trip to Zambezia province in 2016 (Appendix V). Of these, three were identified as Green Sawfish rostra and the rest were from Largetooth Sawfish. The origin (location where sawfish was caught or landed) was known for ten of these rostra. One of the *P. zijsron* rostra came from a sawfish caught by a shrimp trawler based in Beira (Sofala province). The Largetooth Sawfish rostra came from sawfish caught in Nampula (including one caught in the Lurio River, the border between Nampula and Cabo Delgado provinces) and Zambezia provinces. The standard rostral lengths of rostra ranged from 29 cm to 127 cm for *P. pristis* and 91 cm to 110 cm for *P. zijsron*. An additional five rostra known to be of Mozambican origin, all from Largetooth Sawfish, were documented from three museums in Europe and have been included with the historical reports (Appendix II).

## DISCUSSION

This is the first systematic study on the distribution, exploitation and status of sawfishes in Mozambique. Sawfishes are clearly still encountered by artisanal fishers, and 23% of those interviewed had seen or caught a sawfish between 2010 and 2014. In a similar study conducted in Guinea-Bissau, only 12% of artisanal fishers interviewed reported sightings of sawfishes between 2005 and 2012 (*Leeney & Poncelet, 2013*). This suggests that recent catches and encounters with sawfishes are more common in Mozambique, at least in certain areas. Both the Green and Largetooth Sawfish rostra were documented during this study, but rostra from Largetooth Sawfish were considerably more numerous and comprised all of the rostra accompanying recent reports of sawfish encounters. Both species were also formerly present along South Africa's KwaZulu Natal coast, but are now considered to be extinct (*Everett et al., 2015*).

Sawfish catches and observations were stated to have occurred in all coastal provinces of Mozambique, and combined with the historical data, this information suggests that sawfishes were encountered along much of Mozambique's coast in the past, and likely inhabited several major river systems. The presence of Largetooth Sawfish rostra alongside

reports of recent captures in Nampula and Zambezia provinces suggests that sawfishes may persist in both of these regions. Younger interviewees appeared to be less likely to have encountered a sawfish than those in older age classes. Similar patterns have been observed elsewhere (e.g., *Leeney & Poncelet, 2013*) and may be a result of decreasing sawfish abundance and thus declines in sawfish sightings and captures, causing 'shifting baseline syndrome' in which younger generations are less aware of the diversity or abundance of local species in the recent past (*Turvey et al., 2010*). However, this pattern may also be a function of experience, with older fishers having had more opportunity to encounter rare species such as sawfishes during a longer period of interactions with marine and riverine environments.

During his exploration of the Zambezi River, Livingstone noted: '*We never heard of anyone being wounded by this fish, nor, though it goes many hundreds of miles up the river in fresh water, could we learn that it was eaten by the people*' (*Livingstone & Livingstone, 1866*). In contrast, this study documented the use of sawfishes as a source of food, as well as saleable commodities—primarily fins but also the flesh and very occasionally the rostrum—for fishing communities. In none of the study areas did sawfishes appear to be caught frequently (e.g., on a daily or weekly basis), and as such they likely do not provide a regular source of income except perhaps for fishers who may attempt to target sawfishes, along with sharks and guitarfishes, for their fins. Nonetheless, even sporadic sources of income and any catch that can be consumed are valuable to impoverished fishing communities and even the occasional sale of sawfish fins would represent a significant source of cash for an artisanal fisher in Mozambique, as has been documented for the sale of shark and sawfish fins in other western Indian Ocean countries (*Nyingi et al., 2008*; *Cripps et al., 2015*). Surprisingly, a small number of respondents did not appear to be aware of the value of the sawfish's fins and stated that they were usually discarded. It may be the case that some of the more remote villages are not visited by fin collectors and thus there is no market for sawfish fins in those places.

In some parts of Africa, sawfishes are totemic or culturally significant species. This is particularly the case in West Africa, where sawfishes symbolize strength, protection and prosperity (*Robillard & Séret, 2006*). In Guinea-Bissau, sawfishes are an important part of the Bijago culture's traditional ceremonies (*Leeney & Poncelet, 2013*). In Mozambique, however, no such cultural importance was documented. It may be that any traditions or beliefs associated with sawfishes have been forgotten as the species became rarer, or alternatively, they may never have been seen as anything other than a source of food and a potential danger to fishermen in this region.

Sawfishes and guitarfishes, (the latter locally referred to as '*peixe-viola*'), seem to have often been confused by some fishermen, who recognised the shared body shape of these two groups. Extra care was thus taken during this study to ensure that any responses provided by interviewees pertained to sawfishes and not to guitarfishes. However, information collected after the completion of the interviews documented here revealed that saw sharks are bycaught by industrial and semi-industrial trawlers, and that many fishers and fisheries observers could not distinguish between sawfishes and saw sharks. This suggests that at least some of the interviewees who were involved in industrial and semi-industrial fisheries, both fishers and fisheries observers, may have provided information about saw sharks rather than

sawfishes. However, many of the artisanal fishers interviewed reported animals of 4 or 5 m in length, and capture locations included some rivers and mangrove areas. Both these factors suggest that they had indeed encountered sawfishes rather than saw sharks, as the latter are generally under 1.5 m in length and do not inhabit brackish or freshwater habitats (*Ebert & Fowler, 2014*). In addition, sawfish rostra were observed in a number of communities where interviews took place, confirming that sawfishes were caught in those localities, whilst no saw sharks or rostra thereof were ever observed by the author during the study.

Nonetheless, this finding highlights the importance of ensuring that fisheries observers are well-trained and are aware of the existence of both Pristidae and Pristiphoridae in Mozambican waters, and the need to distinguish between these two groups when documenting landings. Saw sharks are not listed as threatened by the IUCN Red List, but little data exist on the levels at which they are bycaught in fisheries in any region. Future studies using interviews to collect LEK on sawfishes in other countries should investigate whether saw sharks occur there and if so, should incorporate this into the structure of the interview by attempting to clarify with interviewees which of the two taxa they have encountered. The latter can be achieved in some cases by showing good-quality photographs of both taxa, and otherwise by collecting as much information as possible on the size and morphology of the animals (e.g., ascertaining whether barbels have been observed on the rostrum, which indicates Pristiophoridae), and the type of habitat in which the animals are caught (rivers and estuaries would indicate Pristidae).

Some of the factors which interviewees perceived as contributing to declines in sawfish abundance do likely reflect the threats faced by sawfishes in Mozambican waters. The findings of this study suggest that sawfishes are caught in both artisanal and industrial fisheries in Mozambique. In the past, directed fisheries for sharks operated in Maputo Bay and Inhambane Bay, targeting coastal and shelf-associated species (*Sousa, Marshall & Smale, 1997*), and these fisheries may well have included sawfishes in their catches. A limited number of gill net fisheries are still licensed to catch elasmobranchs (*Kiszka & Van der Elst, 2015*). Gill nets were the gear type most frequently cited as having resulted in the capture of sawfishes. A considerable bycatch of demersal sharks and rays is known to occur in the prawn trawling industry (*Fennessy, 1994*; *Fennessy & Isaksen, 2007*) and sawfishes are known to be particularly susceptible to capture by this type of fishing activity (*Stobutzki et al., 2002*; *Brewer et al., 2006*; *Simpfendorfer, 2014*). This study collected some evidence to suggest that sawfishes have been bycaught during trawl activities in Mozambican waters (but see caveats, above). Bycatch reduction devices (BRDs) have been tested in prawn trawl fisheries in Mozambique in 2005 and appeared to be effective in reducing bycatch of batoids (*Fennessy & Isaksen, 2007*; *Fennessy et al., 2008*), and national legislation has required the use of Turtle Excluder Devices (TEDs) in trawl fisheries since 2005, but they are not currently used (S Fennessy, pers. comm., 2015). However, a study in northern Australia showed that TEDs reduced only the bycatch of the Narrow Sawfish (*Anoxypristis cuspidata*), whilst other sawfish species are caught when their rostra become entangled in the trawl net forward of the TED (*Brewer et al., 2006*). These and other BRDs currently in use thus appear unlikely to exclude sawfishes from industrial trawl fisheries in Mozambique and innovative research is urgently needed to address this issue.

Bycatch in gill nets is also a critical issue for a range of vulnerable marine species including marine mammals, turtles and elasmobranchs (e.g., *Pusineri et al., 2013*; *Kiszka & Van der Elst, 2015*). Gill nets have been shown to be a key threat to sawfishes in a number of other regions (e.g., *Hossain et al., 2013*; *Giglio et al., 2016*). A recent visit by the author to Zambezia province revealed that local authorities are providing gill nets and motors to artisanal fishers in certain areas, to encourage them to stop using beach seine gear and to move their fishing activities from shallow coastal and estuarine environments to offshore areas (R Leeney, pers. obs., 2016). This initiative has been developed in order to reduce the catch of juvenile fishes in known nursery habitats. IIP representatives assured the author that despite the non-selective nature of gill nets, fishers are aware that turtles and other protected species must be released. However, fishers in these areas know the value of shark fins and have access to markets for them and it thus seems likely that increased gill netting will result in increased levels of elasmobranch landings. Shark fisheries are known to have increased in size and extent in Mozambique in recent decades, driven by the demand for shark fins, and those of shark-like batoids such as guitarfishes and sawfishes, for export (*Pierce et al., 2008*). This industry has likely had an impact on sawfish populations, even if sawfishes have not been directly targeted by fishers for their fins, as most of the fishers interviewed for this study were aware that sawfish fins had some value. A wide range of sale prices for sawfish fins were provided by interviewees, which may be due to variability in the prices offered by buyers, or the significant decline in the value of fins in more recent years. Nonetheless, when considered relative to Mozambique's (2015) annual Gross National Income per capita of US$580,[3] it is apparent that the fins of sawfishes, guitarfishes and sharks are a valuable commodity for fishers in rural communities and they are thus unlikely, at present, to be motivated to release accidentally-caught sawfishes alive.

Exploitation of natural resources is often conducted without consideration of the potential negative effects on coastal and riverine biota, particularly in developing countries eager to grow their economies. Freshwater and estuarine environments are known to provide critical habitat for both the Largetooth and Green sawfishes (e.g., *Morgan et al., 2015*; *Whitty et al., 2009*), and freshwater and euryhaline elasmobranchs suffer from an elevated exposure to threats in their more restricted habitats (*Lucifora et al., 2015*). The degradation of these habitats is caused by activities such as mining, mangrove deforestation, dam construction on major rivers, increases in pollution or toxic influxes; upstream deforestation and increased sediment run-off causing siltation of estuaries (*UNEP, 2007*). Some of the most extensive mangrove communities on the East African coast are found in the Zambezi Delta, which is one of the most diverse and productive river delta systems in the world and is designated as a Wetland of International Importance. The Zambezi Delta faces considerable threats including the overuse of resources due to human pressure, pollution, deforestation and reduced water flows caused by droughts and water abstraction (*Beilfuss & Brown, 2010*; *Schuyt, 2005*). Likewise, heavy sands mining operations for ilmenite, zircon and rutile are taking place at several sites in coastal Nampula province (*ITIE Moçambique, 2015*; *Coastal and Environmental Services, 2000*), and a similar operation has been proposed for a site at the mouth of the Zambezi River (S Nazerali, pers. comm., 2015). Such changes have the potential to negatively impact sawfishes through changes to critical freshwater,

[3]GNI per capita (formerly GNP per capita) is the gross national income, converted to US dollars using the World Bank Atlas method, divided by the midyear population. http://data.worldbank.org/country/mozambique.

estuarine and coastal habitats (*Kyne & Moore, 2014*). A high degree of female reproductive philopatry has been documented for Largetooth Sawfish (*Feutry et al., 2015*), which may imply that if habitat degradation causes a sawfish population to be excluded from an area, that population may not be able to relocate to a less impacted habitat. The identification and protection of critical habitats for sawfishes is thus likely to be a key strategy in the conservation of this taxon (*Kyne & Moore, 2014*).

Sawfishes are amongst the largest predatory fishes in estuaries and shallow coastal waters and, when populations were larger, likely had significant influence throughout their range, both directly and indirectly, on tropical and sub-tropical fish communities (*Dulvy et al., 2014*). It is difficult to assess the implications of the reduction and extinction of sawfish populations in African waters, given the many other pressures these ecosystems also face, but the depletion in general of top predators such as sharks likely impacts on the dynamics of estuarine, coastal and pelagic ecosystems (*Heithaus et al., 2008*).

Sawfishes are not currently protected in Mozambican waters. The white shark *Carcharodon carcharius* is the only elasmobranch species for which all take is prohibited (Article 14, Decree n. 51/99, 31 August, Regulation of Fishing and Recreational Sports). However, Mozambique is a party to the Convention on International Trade in Endangered Species of Wild Flora and Fauna (CITES), which lists all five species of sawfishes in Appendix I. As such, international commercial trade in sawfish parts should be prohibited, but information provided during this study suggests that the fins of sawfishes are still sold to collectors for export to Asia. The formal development of a National Strategy for Sawfish Conservation is strongly recommended, but activities focused at a local level and involving community members from key sawfish habitats will likely be the most effective means of protecting sawfishes in Mozambique.

Social issues and contexts such as human well-being, values and cultural norms determine the potential for changes in human behaviour and are therefore intrinsically linked to the opportunities and constraints for successful conservation action (*Cowling & Wilhelm-Rechmann, 2007*). The data collected during this baseline study suggest that sawfishes persist in Mozambique. It is now essential to verify the specific habitats used by sawfishes; population sizes and local threats, in order to develop appropriate conservation and management plans for sawfishes in Mozambican waters. However, Mozambique ranks 178th out of 187 countries on both the Human Development Index[4] and the Income Gini Coefficient[5]; its artisanal and subsistence fishing communities are amongst the poorest in the country. The findings of this study suggest that sawfishes provide at least an occasional source of food for fishers in some areas, and may also be a significant, if opportunistic, source of cash through the sale of their fins. Thus, if sawfish conservation measures developed for sawfish habitats where artisanal fishing takes place are to be effective, they will have to incorporate diverse social data including the contribution that sawfishes may make to a fisher's livelihood, and should offer incentives or alternative means of generating income. Future plans for the conservation of sawfishes and their habitats must be developed in close collaboration with the resource-dependent people who rely upon sawfishes and their habitats for their survival. Without immediate action, these remnant

[4]The Human Development Index is a composite index calculated by the United Nations Development Program, measuring average achievement in three basic dimensions of human development–a long and healthy life, knowledge and a decent standard of living. http://hdr.undp.org/en/content/table-1-human-development-index-and-its-components.

[5]Measure of the deviation of the distribution of income among individuals or households within a country from a perfectly equal distribution (*World Bank, 2013*). http://hdr.undp.org/en/content/income-gini-coefficient.

sawfish populations are likely to decline rapidly or become extinct, as have those in other parts of Africa (e.g., *Everett et al., 2015*; *Leeney & Downing, 2015*; *Leeney & Poncelet, 2013*).

## ACKNOWLEDGEMENTS

Transport to interview sites was provided by IIP in the northern Zambezi province and by the WWF-CARE team in Nampula Province. Special thanks to the following people at IIP in Mozambique: Paula Santana Afonso, Osvaldo Chacate, Alice Inaçio and Daniel Mualeque. I am grateful to the many individuals who acted as interviewers or translators: Martinho Padeira, Eurico Morais, Afonso Munduze, Ussene Muarecha, Muhede Ali, Luis Anselmo, Eduardo Avene, Acurçio Cumbane, José Vilankulo, Alexandre Thuzine, Francisco Zivane, Rui Mutombene, Isaias Tembe, Sujado Zivane and Abu Junior. Thanks also to Karen Allen (EWT), Peter Bartsch (Museum für Naturkunde), Peter Bechtel, Libby Bowles, Inês Carvalho, Simon Chitsenga (WWF-CARE), Marion Duffin (Museu de História Natural, Maputo), Nick Dulvy, Nicole Helgason, Johanna Kapp (Museum für Naturkunde), Robert and Niamh Leeney, James Maclaine (BMNH), Andrea Marshall (MMF), Alec Moore, Nataniel Anildo Naftal (Eduardo Mondlane University), Simon Pierce (MMF), Ernesto Poiosse (IDPPE), Clare Prebble (MMF), Ana Cristina Rufino (Museu da Ciência, Universidade de Coimbra), Chris Scarffe, Chris Smith, Elizabeth Stephenson (MCAF), Rudy van der Elst, Willie van Duyvenbode, Gail Wearne and Simon Wearne. Many thanks to Monica Clerio for contributing several historical records of sawfishes in Mozambican waters from the International Sawfish Encounter Database. This manuscript has been improved by comments from the journal editor and two anonymous reviewers.

### Funding

This study was funded by the New England Aquarium's Marine Conservation Action Fund, the Swiss Shark Foundation and the Rufford Small Grants Foundation. Ruth H. Leeney is funded by NOAA Fisheries grant NA15NMF4690193 and the Leonardo Dicaprio Foundation. The funders had no role in study design, data collection and analysis, decision to publish, or preparation of the manuscript.

### Grant Disclosures

The following grant information was disclosed by the author:
New England Aquarium's Marine Conservation Action Fund.
Swiss Shark Foundation.
Rufford Small Grants Foundation.
NOAA Fisheries: NA15NMF4690193.
Leonardo Dicaprio Foundation.

### Competing Interests

The author is Director of the organisation 'Protect Africa's Sawfishes', which collects baseline data on sawfish populations throughout the developing world.

## Author Contributions

- Ruth H. Leeney conceived and designed the experiments, performed the experiments, analyzed the data, wrote the paper, prepared figures and/or tables, reviewed drafts of the paper.

## Human Ethics

The following information was supplied relating to ethical approvals (i.e., approving body and any reference numbers):

Written consent was not received from interviewees for each interview to go ahead, as many interview participants were unable to read. Permission to conduct the interview was requested verbally from each interviewee prior to starting the interview.

## Data Availability

All sawfish records collected during this study were provided for peer review and to the International Sawfish Encounter Database (managed by the Florida Museum of Natural History; https://www.flmnh.ufl.edu/fish/sawfish/ised/). Due to the sensitive nature of the information, it cannot be made public.

## Supplemental Information

Supplemental information for this article can be found online at http://dx.doi.org/10.7717/peerj.2950#supplemental-information.

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
