# Peer review of "Are sawfishes still present in Mozambique? A baseline ecological study"

_PeerJ, doi:10.7717/peerj.2950_

## Round 0.1 · original submission · Major Revisions

The two reviewers have raise a few issues that I think should be addressed and that would improve the manuscript. Regarding the summary spreadsheet that documents the interviewees responses, it is rather unclear which would make it difficult for someone repeating the analysis to come to the same results; to a certain extent this comes through in the manuscript. It would be good if the spreadsheet could be better structured/organised.

Reviewer 1 ·

Basic reporting

The topic of the paper is essentially very interesting and timely given increasing calls for baseline data on sawfishes throughout their historical range (see Harrison & Dulvy 2014). The paper is largely well written and conforms to the reporting standard for PeerJ. With regards to content the introduction provides a broad overview of the presence of sawfishes in Africa, and of the status of the fishery sector in Mozambique. The latter, however, could do with some clarifications and additions from the sea around us project. The methods and results are largely clear, though the figures would benefit from edits such as producing multi-panel plots rather than several individual figures. The discussion provides a good overview of the results, however, some of the assertions in the discussion are speculative and I feel could be improved by additional analyses – especially when discussing conservation interventions and activities (see below). In terms of raw data - the authors have provided a summary of the interview results, as well as data on historical records and sawfish observations gathered during the study.

Experimental design

This study is based on a questionnaire, which in its current form is more a rapid assessment as it is based on 14 questions (note: Q9 and Q13 are very similar). I feel that with some extra consideration this questionnaire could have been greatly improved to reduce possible errors associated with misidentification of sawfishes and also to obtain more detailed information that would benefit future efforts in the region. For instance: (1) where were sawfishes encountered – i.e. distance from shore or landing site (2) which habitats were they encountered in; (3) what depths were they encountered; (4) what gear was used – including information on mesh size, length, height and/or line length, number of hooks, and typical soak times; and (5) the seasonality of captures. I appreciate this requires fishers to recall their efforts but this data would provide a lot more context to the threats and pressures facing these enigmatic and critically endangered species. The current findings thus provide little more information than that sawfishes have been observed in Mozambican waters in the past and up until present in fisheries. If as scientists, practitioners, managers we want to implement changes we need to better understand more about where and how they were caught, as well as about the demographics of those who are involved in their capture.

Furthermore, given the similar characteristics/morphometrics of several species in this region (sawfish, sawshark, guitarfish/rays) the questionnaire would have benefitted from an assessment of fishers ability to correctly identify sawfish – this is clearly an issue as highlighted by the author (I appreciate their honesty but it makes me question the validity of their results – see comments below). I feel that the structure of the methods and results could also be refined to flow more logically to address the following topics: demographics including profession, records of captures (by sector), gear types used, fishers perceptions of changes in abundance/captures, distribution of captures, and end use (i.e. sale, consumption etc). I am also unsure about where best to place the historical data – the reporting of these results seems a bit awkward in its present location – I have two broad views: (1) this data would be better placed in the introduction to provide more context/background to presence of sawfish in Mozambique and thus the need to further investigate its current status; or (2) at the end of the results to compare historical data to more recent data derived from the authors surveys and observations (e.g. species, size etc).

Finally, it is clear from the results that older persons have typically observed sawfishes compared to the younger generation, yet the results show that observations of sawfishes have increased since the 1960s these results seem at odds to me – can the author explain these findings? Thinking out aloud this could either imply (1) that with changes in fishing practices such as the increasing adoption of motors (meaning that fishers operate in deeper waters offshore) respondents are increasingly capturing sawfish in new areas; (2) respondents are misidentifying sawsharks as sawfish given the issues the author highlights; or (3) that older fishers operate differently - untangling these differences is what is of greatest importance. Therefore I feel that some statistical analyses to identity differences (e.g. age, profession, number of years fishing, gear type, location of site etc) between those who have captured and observed sawfishes and those who havent would be useful for future conservation efforts and education/awareness programs. Whilst I appreciate the current data is limited (see concerns about the questionnaire above) this would still prove useful - after all how can we implement behaviour change or implement effective management strategies if we don’t target our efforts at the correct groups or adopt the right measures?

Validity of the findings

As indicated above I feel that more could me made of the results in the discussion - by this I mean a better understanding of the differences in those who are more likely to capture sawfishes. This would allow a more informative discussion around the types of fishers, gears, locations and age groups that should be targeted to reduce pressures on sawfishes, and to ensure more targeted research efforts.

Additional comments

Introduction:
1. Line 23: Include common names for Pristis pristis and P. zijsron
2. Line 33: I feel that the closing sentence of the abstract could be more informative. For instance, we also need to better understand the spatiotemporal distribution of small-scale fisheries activity and effort and their overlap with critical habitats to better inform future management strategies and conservation efforts.
3. Line 34: Bold statement to say that we need to sensitise fishers to the need to release sawfishes alive – I think the statement should address more broadly the need to implement education and awareness programs that are targeted at fisheries groups associated with capture of these species (which in Mozambique are X,Y AND Z as identified from statistical analyses described above).
4. Lines 53 – 54: The toothed rostrum and demersal habits of all sawfish species make them extremely susceptible to capture in gillnets and demersal trawl nets (Simpfendorfer 2013; Kyne et al. 2013) – This could be more clearly stated. For example, ‘The toothed rostrum and demersal habits of all sawfish species make them extremely susceptible to capture as they occupy similar habitats and depths to those utilised by small-scale fisheries which typically deploy gillnets and demersal trawl nets’.
5. Lines 68 - 69: How important is this sector for employment - see Teh and Sumaila (2013)
6. Lines 72 – 77: Several areas of Mozambique are mentioned here – please provide reference to Figure 1 so the readers have something to refer to, to identify their location.
7. Lines 82 – 86: This statement is not clear what do you mean without vessels and working on vessels? Also could you provide a map showing the relative distribution of these vessels across the 1,586 centres of fishing activity to provide more background to the fishery sector.
8. Lines 86 – 89: The following reference illustrates quite nicely the scale of migration that is undertaken in some fisheries in Africa: Human migration and marine protected areas: Insights from Vezo fishers in Madagascar
http://www.sciencedirect.com/science/article/pii/S0016718516300525
9. Line 89: ‘Some specialised artisanal fishers target sharks (RH Leeney pers. obs.)…’ – for fins, meat, both?
10. Lines 101 – 102: The outputs/value/contributon of this research could be more explicit rather than stating recommendations are made.
Methods:
11. Lines 109 – 117: Refer to figure 1 as lots of locations are named/identified.
12. Lines 112 – 114: Why were these areas selected - based on evidence / other studies or that it is an area that provides suitable habitat?
13. Lines 119 – 184: See comments above about structure of methods suggested in experimental design.
14. Line 133: Why did you include data from IIP – could you have influenced their results given that you used this organisation to help collect data and would of trained them to help assist you?
15. Line 168 – 184: See comments above about additional analyses described in experimental design that would you to broadly identity differences between those who capture sawfishes and those who don’t (e.g. age, profession, number of years fishing, gear type, location of site etc)
16. Line 179: Given the similar characteristics/morphometrics of several species in this region (sawfish, sawshark, guitarfish/rays) the questionnaire would have benefitted from an assessment of fishers ability to correctly identify sawfish – this is clearly an issue as highlighted by the author (I appreciate their honesty but it makes me question the validity of their results. Some sort of assessment of the potential error in identification would help address these issues or allow you to caveat your findings – e.g. a supplemental analysis that investigated the ability of fishers to correctly identify species revealed that x% of surveyed fishers could correctly identify the difference between a sawfish and other species.
Results:
17. Lines 190 - 203: historical records of sawfishes – see comments above regarding restructuring of the result section and the moving this information to the introduction to provide more context for the importance of this study. These results would also be better presented as a figure showing the number (or proportion) of historical records at each reported location, which could be referred to in the introduction.
18. Line 209: Regarding the group of fishers – how was this data used and presented.
19. Lines 210 – 212: Present percentage distribution of gear use in the text and refer to Figure 4. In addition, separate gear use by sector in figure 4 so that reader can untangle which gears are more associated with capture of sawfishes in each sector.
20. Line 217: Avoid starting sentence with Figure 2 shows.
21. Line 217: delete ‘who provided their age’
22. Line 219: ‘The majority of respondents…’ – provide descriptive statistics here. Also you fail to refer to Figure 2 in the text when talking about age distribution of respondents.
23. Line 228 – 234: Local names – whilst interesting these results are not referred to elsewhere in the paper or discussion and so would be best placed in supporting information and referred to on Line 415 of the discussion – where you mention local names.
24. Line 239: There is some discrepancies in sample sizes throughout for instance on Line 239 the author states ‘Of 165 interviewees….’ yet on line 210 the author states ‘Of the 200 interviewees, 160 were artisanal fishers…’. The description of sample sizes for respondents needs to be clearer throughout. I repeatedly found myself trying to understand where the values come from, and having to refer back to previous paragraphs as the values change so often.
25. Lines 240 – 241: The dates of respondents’ most recent sawfish sightings and 241 captures ranged from 1968 to ‘last week’ (indicating July 2014) – how confident are you that these are not sawsharks given the issues you highlight?
26. Line 241: Don’t start a sentence with a percentage - ‘29% of respondents said…’
27. Line 247 – 248: ‘Several recent observations were also reported in Inhambane province, but these may have all been of the same one animal’ - What evidence is there for this - why do you infer that it could be the same animal?
28. Line 258 ‘….some kind of gillnet’ - what do you mean here, be more explicit or at least describe the variation in this gear type.
29. Lines 260 – 261: ‘Sawfishes also appear to have been caught by a diverse range of other gear types, but in far fewer instances.’ – such as which gears?
30. Line 265 – 266: This sentence could be better phrased. For example. "Of 125 respondents who had observed sawfishes 30% (n = 37) stated that they had only observed them once."
31. Line 266 – 267: These respondents were not asked whether they had noted a change in sawfish abundance over time. – Why is this statement here when the following sentence pasted below talks about fishers’ perceptions and local declines.
32. Lines 267 – 270: ‘In general, the question was not well understood by interviewees and only 32 individuals provided a response: the majority (n=27) stated that sawfish numbers locally had declined over the course of their lifetime, or suggested possible factors affecting a decrease in abundance from which an observed decline could be inferred.’ – the latter part of this sentence from ‘..or suggested..’ onwards is not clear.
33. Lines 272 – 278: Could the findings of these results be plotted as a freq. histogram (i.e. fishers perceptions of declines in sawfishes) with results categorised (i.e. increase in boats, fishers, changes in fishing practices etc).
34. Lines 281 – 299: I appreciate the sensitivity of these results but also where are these data going to be reported if not here. How can NGOs, local and national institutions implement management actions or assess behaviour change if this data is not accessible. I think there could be a compromise - such as reporting at a coarse scale (i.e. assign counts of observations to a coarse scale/resolution vector grid that could be mapped).
35. Lines 302 – 323: I feel that the results of cultural & socioeconomic importance would be better plotted and clearer if plotted as a frequency histogram (per figs 1, 2 and 3) with categorised responses (e.g. consumption, sale, etc).
36. Lines 310 – 312: ‘The sale prices quoted for sawfish fins were far greater than those for the meat, and the latter appeared to be sold only for local consumption’ – what are these prices? Is there a reason they are not reported?
37. Line 322: delete ‘from’ ‘…used as hooks for hanging clothes from,…’
38. Lines 345 - 364: Sawfish Rostra results data could be added to a two part map discussed in comment 34 above - (A) historical data; and (B) reported by fishers. This would highlight areas where there is agreement in the presence of sawfish, and or where species were once present and no longer observed or where still present despite changes in fisheries.
Discussion:
39. Lines 372 – 374: ‘When compared with a similar study conducted in Guinea-Bissau, where only 12% of respondents reported sightings of sawfishes within the period 2005-2012 (Leeney & Poncelet 2013), this suggests that sawfishes are more commonly encountered in Mozambique’ – Did this study use the same questionnaire, are the sample sizes similar, did they engage with similar sectors – how confident are you that the results of these studies are directly comparable. I would assume the fisheries sectors differ.
40. Lines 416 – 418: ‘Extra care was thus taken during this study to ensure that any responses provided by interviewees pertained to sawfishes and not to guitarfishes’ – I don’t agree with this statement, there is no assessment of respondents abilities to correctly identify sawfishes.
41. Line 435: FEK? What does this mean?
42. Lines 453 – 454: ‘….and this study collected some evidence to suggest that sawfishes have been bycaught during trawl activities in Mozambican waters…’ – change bycaught to captured.
43. Lines 454 – 455: ‘Bycatch reduction devices have been tested in prawn trawl fisheries in Mozambique in 2005 and appeared to be effective…’ – effective for which species and do any of these species have similar life history characteristics, habitat requirements or behaviours for sawfish that mean it might also be beneficial?
44. Lines 472 - 473: ‘Shark fisheries are known to have increased in size and extent in Mozambique in recent decades,…’ add descriptive statistics from the sea around us project catch database to reinforce this point.
45. Lines 474 – 477: ‘This industry has likely had an impact on sawfish populations, even if sawfishes have not been directly targeted by fishers for their fins, as most of the fishers interviewed for this study were aware that sawfish fins had some value, making them less inclined to release any accidentally-caught sawfish.’ – This statement is speculative as the author did not directly address this topic in there questionnaire.
46. Lines 510 – 512: ‘The formal development of a National Strategy for Sawfish Conservation is strongly recommended, but activities focused at a local level and involving community members from key sawfish habitats will likely be the most effective means of protecting sawfishes in Mozambique’ – This is where untangling the differences in those who capture or have captured sawfish compared to those who haven’t is important (see comments for experimental design and additional analyses detailed in comment 15).
47. Line 517: replace persist with present.
Figures:
48. Figure 1 needs an inset map to show location of (a) and (b).
49. Figure 2 would benefit from a kernel density line to nicely illustrate the distribution of the age groups that have observed sawfishes and those who haven’t. See http://www.r-bloggers.com/exploratory-data-analysis-combining-histograms-and-density-plots-to-examine-the-distribution-of-the-ozone-pollution-data-from-new-york-in-r/ which provides a simple example to illustrate.
50. Figure 3 - Standardis the axis labels - 1960s, 70s. 80s, 90s, 2000s, 2010-present.
51. Figure 3 - Can you weight bars by number of respondents.
52. Figure 4 – present results by sector (i.e. artisanal, industrial)
53. Figures 2 – 4 combine into one multi-panel figure (A, B, C) and provide sample sizes for each within the plot (n = ?).

Reviewer 2 ·

Basic reporting

Submission adheres to PeerJ policies. Article is written in good English, and except for some minor comments listed below, text is very well developed in all segments of the manuscript. All figures and tables are meaningful and relevant to the manuscript contents. Author has also submitted raw data - I am not sure if it was also uploaded in an online repository, if not that should be also completed.

Experimental design

Study seems valid and well designed. Surveys and data collection were well organized and the whole coast of the country was well covered by the study. Study provided relevant new knowledge, which was adequately presented in the manuscript. Ethical standards seem to have been followed appropriately.

Validity of the findings

Data that were obtained and presented in this study seem to be robust and sound. With the exception of some minor comments given below, discussion and conclusions are well formed and presented.

Additional comments

In the study titled "Are sawfishes still present in Mozambique? A baseline ecological study" (#10357), author has made an extensive survey along the coast of Mozambique, interviewing fishermen and other stakeholders, and collecting other sources of information, in order to obtain information on the presence and status of sawfishes in this area. Given the highly endangered status of this species group worldwide and gaps in knowledge on their populations in this region, presented study provided relevant new knowledge. Study and the manuscript are well developed and, given that author takes into consideration minor comments provided below, I would strongly support publication of the manuscript.

1. MS title - delete the dot at the end of the title

2. Sawfish family (Pristidae) should be presented when the group is mentioned for the first time, both in the Abstract (Line 11) and Introduction (Line 40), the same as it was done for guitarfishes (Line

3. Acronyms should be also accompanied with the name written in full at its first mention in the text: IUCN (Line 50), EEZ (Line 111), SRW and SRL (Lines 159-160), FEK (Line 435), TED (Line 456)

4. IDPPE full-name is listed twice in the text (Lines 83 and 136-137), so the second instance should be deleted.

5. Line 48 and elsewhere - species names such as Largetooth Sawfish should not start with capital letters

6. Although it is not much important, author uses interchangeably terms fisher and fisherman. Related to this topic: Branch, T. A., & Kleiber, D. (2016). Should we call them fishers or fishermen? Fish and Fisheries.

7. Section on Interview surveys in Methods is somewhat confusing. Although author also interviewed sellers and monitoring staff, in Line 131 it is stated that only "artisanal, semi-industrial and industrial fishers were interviewed". Furthermore, this section includes paragraphs with the description of visits to natural history museums (Line 156) and fishing and dive operators (Line 163) which do not really represent interviews, especially since they were not included in the analyzed sample, presented in table 1. Types of respondents should be listed at the beginning of this section, and the interviews that were not used as a part of the analyzed sample should be described separately.

8. Section on Analysis in Methods does not really present analytical approach. Study is mainly descriptive, so there were no statistical methods used. Nevertheless, author should use this section to describe main topics that were researched, questions that were addressed - e.g. main outcomes that were targeted by the survey, by collection of data on rostra, etc. Most of this section currently contains discussion about the reliability of responses, due to potential mixing of sawfish and saw sharks by respondents. That paragraph can be also moved to a separate subsection, titled for instance "caveats".

9. Author should also address another potential caveat, either in Methods or in Discussion section, namely whether respondents could have been motivated to withold some of the requested information, either to protect their activities or just as a matter of precaution - this often happens in interviews with fishermen potentially engaging in illegal activities. I am not sure how much that was possibility in this study, but it should be addressed nevertheless.

10. Lines 202-203 - "This suggests...that sawfishes were abundant in Mozambican waters" - such conclusion on historic abundance levels can not be drawn with certainty from presented historic records. Maybe "abundant" should be replaced with "relatively abundant".

11. Lines 165-166 - it would be clearer if "full sawfish, dead" and "full sawfish, alive" would be replaced with "whole body" and "alive specimen"

12. Line 248 - replace "one animal" with "individual"

13. Line 302 - replace "&" with "and"

14. Discussion is very nicely developed.

15. Line 415 - family for guitarfishes is already given in Line 233.

16. Line 497 - delete "in" before the word "fishes"

17. Line 526 - replace "peoples" with "people"

18. Line 527 - I do not agree with the statement that "sawfish populations are likely to decline rapidly". Given the lack of knowledge presented in this paper, it is possible that the major decline has already occurred, so the next step is actually local extinction. I would after "decline rapidly" add "or become extinct"

19. Figure 1 - it would be useful if names of all provinces and bays mentioned in the text would be added in the map.

20. Figure 4 - in figure caption, "Cumba refers to a type of dragged gill net" should be deleted, as the data on Cumba type net are not shown in the figure.


References:

21. Reference Leeney is cited as published in 2015 in Line 46, while in the reference list it was presented as published in 2014 (Line 642), in case it is the same reference

22. Reference in Line 64 is cited with first author's last name "Fenrnandez-Carvalho" and as published in 2013, while in the reference list it was presented with first author's last name "Fernandez-Carvalho" and as published in 2014 (Line 610), in case it is the same reference

23. Reference Whitty et al. (Line 161) has 2013 as year of publishing in the text (also in Appendix IV caption), and 2014 in the reference list (Line 723).

24. Reference van der Elst et al. 2010, cited in Line 79, is missing from the reference list.

25. Order of references in the list should be checked - papers with the same first author and more than two coauthors should be ordered chronologically, not alphabetically.

26. Reference Dick-Read 2005 is not presented in the manuscript, only in Appendix I, so it should be listed there and not in the reference list (Line 574).

27. Line 601 - publication title should not be written in capital letters

28. Line 669 - "Phillips et al. 2011" at the beginning of the line should be deleted.

29. Appendices should be each accompanied wiith their own reference lists when needed

---

## Round 0.2 · Minor Revisions

Thanks for submitting the corrected version of your manuscript, this is much improved. I've only managed to get a review from one of the previous reviewers and not wanting to bring in another reviewer who hasn't followed the manuscript through (I know how annoying this can be when it happens!) I have looked at the paper and agree that only minor corrections are needed based on the reviewer's comments.

Reviewer 2 ·

Basic reporting

No Comments

Experimental design

No Comments

Validity of the findings

No Comments

Additional comments

Author addressed adequately all the comments made by both reviewers, and the revision resulted in improved overall quality of the manuscript. As I commented before, the topic of the study is highly relevant due to the status of the studied species group and a lack of knowledge that the author attempted to address, while study seems to be valid and the manuscript is well developed.

As a result, I find the manuscript acceptable for publication in its present form, following a few minor corrections listed here:

1) Line 333 ("The majority (n=27) stated or inferred" - replace "inferred" with "indirectly indicated" or "implicitly indicated" or something in that sense

2) Line 368-369 ("whilst 108 individuals provided a total of 234 responses regarding the uses of various parts of sawfishes") - this is confusing, it would be better to state simply "whilst 108 individuals suggested different uses of sawfishes and its parts" - it is understandable that they often suggested more than a single use, and it is explained as well in the Fig. 5 legend

3) Please check again whether all the numbers in the manuscript match regarding the respondent numbers for different questions - for instance, in Line 367 it is stated that 17 respondents did not provide any answer regarding the use of sawfishes, while according to Fig. 5 legend this number was 16 (which seems incorrect as, according to Line 368, 108 did provide answers and n=125)

4) There is a repetition in Lines 390-391 ("Two interviewees mentioned extracting oil from
sawfishes for use as a sealant or waterproofing agent for wooden pirogues, or for cooking") and in Lines 393-394 ("as a source of oil for cooking (1 respondent) or as a waterproofing agent for fishing boats (1 respondent);")

5) Line 518 - word "Threatened" should not be capitalized as it does not represent a true IUCN Red List category (i.e. there are categories "Near Threatened", "Vulnerable", "Endangered", etc., but not "Threatened")

6) Reference list should be re-checked for correct ordering of references - i.e. it is a common practice that, for the papers with the same first author, one lists first single-author papers, then two-author papers (ordered alphabetically, and then chronologically when both authors are the same), and afterwards all with three or more authors (ordered chronologically) - e.g. reference Fennessy and Everett 2015 should be listed after Fennessy 1994

7) In Fig. 2, if possible, "no ans" should be written in full, "no answer", or maybe "none"

8) In Fig. 4, it would be less confusing if the category "other" would be the last one (i.e. actually the first shown, on the top of the graph)

9) In Fig. 5, replace name of the category "sold" with "sold whole", and "eat" with either "eaten" or "consumed"

10) In the second row of Table 1, replace "industrial trawler/ semi-ind" with "industrial / semi-industrial"

---

## Round 0.3 · accepted · Accept

Many thanks for making the corrections which I'm now happy with. I look forward to seeing it published.